# Gamma-Delta T-Cell Phenotype and Function in DAA-Treated HIV-HCV Co-Infected and HCV-Mono-Infected Subjects

**DOI:** 10.3390/v14081594

**Published:** 2022-07-22

**Authors:** Valeria Bono, Camilla Tincati, Lorena Van Den Bogaart, Elvira Stefania Cannizzo, Roberta Rovito, Matteo Augello, Anna De Bona, Antonella D’Arminio Monforte, Laura Milazzo, Giulia Marchetti

**Affiliations:** 1Clinic of Infectious Diseases and Tropical Medicine, San Paolo Hospital, ASST Santi Paolo e Carlo, Department of Health Sciences, University of Milan, Via A. di Rudinì, 8, 20142 Milan, Italy; valeria.bono@unimi.it (V.B.); stefania.cannizzo@gmail.com (E.S.C.); roberta.rovito@unimi.it (R.R.); matteo.augello@unimi.it (M.A.); anna.debona@asst-santipaolocarlo.it (A.D.B.); antonella.darminio@unimi.it (A.D.M.); giulia.marchetti@unimi.it (G.M.); 2III Division of Infectious Diseases, Luigi Sacco Hospital, ASST Fatebenefratelli Sacco, Department of Clinical and Biomedical Sciences, University of Milan, Via G. B. Grassi 74, 20157 Milan, Italy; lorena.vandenbogaart@unimi.it (L.V.D.B.); laura.milazzo@asst-fbf-sacco.it (L.M.)

**Keywords:** γδ T-cells, Th17-cells, Treg cells, B-cells, HCV, HIV, DAA, liver damage

## Abstract

HIV-HCV co-infected subjects are at risk of liver fibrosis which may be linked to immune imbalances. Direct-acting antivirals (DAAs) represent the mainstay of HCV treatment in co-infected individuals, yet their effects on immune cell populations playing a role in fibrogenesis is unknown. We assessed γδ T-cell phenotype and function, Treg and Th17 frequencies, as well as γ-globulins and B-cell activation in 47 HIV-HCV co-infected and 35 HCV mono-infected individuals prior to and following DAA treatment (SVR12). Γδ T-cell activation decreased in both groups yet persisted at higher levels in the HIV-HCV co-infected subjects. No differences were registered in terms of γδT-cell function. Of note, the Vδ2/Th17 ratio, inversely linked to liver damage, increased significantly in the two groups upon treatment, yet a negative correlation between the Vδ2/Th17 ratio and liver function enzymes was found in the co-infected subjects alone. B-cell activation and γ-globulin levels decreased in both settings, yet B-cell activation remained higher in the HIV-HCV co-infected individuals. In HIV-HCV co-infected and HCV mono-infected participants, the effect of DAA was limited to γδ T- and B-cell activation as well as γ-globulin concentrations and the Vδ2/Th17 ratio, with no changes in γδ T-cell function and Treg frequencies. Importantly, γδ T- and B-cell activation remained at higher levels in the co-infected individuals than in those with HCV mono-infection alone. The persistence of such alterations within these cell subsets may be associated with the risk of hepatic and extrahepatic complications.

## 1. Introduction

HIV-HCV co-infected subjects are at risk of accelerated liver fibrosis [1] and disease progression. The underlying causes are multifactorial and include, yet are not limited to, life-style habits, uncontrolled HIV replication, and the use of combination antiretroviral therapy (cART) [1].

Immune skewing in HIV-HCV co-infected individuals may represent a key element of hepatic fibrosis [2]. Indeed, a solid bulk of the literature has demonstrated that enduring immune abnormalities, despite effective antiviral therapy against either HIV or HCV, may be a pathogenic feature of hepatic disease progression in HIV infection [3,4,5,6,7,8,9,10].

Direct-acting antivirals (DAAs) currently represent the mainstay of anti-HCV treatment in HCV and HIV-HCV co-infected populations, given their ease of administration, clinical efficacy, and tolerability. Further, in both clinical settings, DAA treatment is able to lower the expression of activation and senescence markers on T-cells as well as the levels of inflammatory markers in the peripheral blood [11,12]. However, in HCV mono-infection, DAAs fail to restore the frequency and function of Th17, Treg [13], and γδT-cells [14,15] which are known to play a pivotal role in fibrogenesis [16,17,18,19,20] and, in the case of Tregs, may also contribute to the pathogenesis of extrahepatic manifestations in chronic HCV disease [21]. Scant data are available on the effects of DAA treatment on Treg and γδT-cells subsets in HIV-HCV co-infection. Indeed, with the exception of one study reporting impaired IFN-γ production in γδT-cells in treated HIV-HCV co-infection [22], the impact of DAA therapy on these cell populations in the co-infected individuals is largely unknown.

To bridge this gap, we investigated the changes in γδT-cells activation and function as well as Th17 and Treg cell frequencies, γ-globulins, and B-cell activation in a cohort of DAA-treated HIV-HCV co-infected and HCV mono-infected individuals.

## 2. Methods

### 2.1. Study Population

We enrolled HIV-HCV co-infected and HCV mono-infected subjects that were undergoing treatment with DAAs. Both groups were studied prior to (baseline, BL) and at 12 weeks after the end of treatment (EoT), corresponding to a Sustained Virological Response (SVR12). The co-infected HIV-HCV subjects were all virally suppressed for HIV on cART.

The local Institutional Review Board approved the study, and written signed informed consent was obtained from all the patients.

### 2.2. Immune Studies

Peripheral blood mononuclear cells (PBMCs) were isolated from EDTA tube-collected venous blood by means of Ficoll-Histopaque (Biocoll, BIOSPA, Milano, Italia), cryopreserved in fetal bovine serum (EuroClone) with 10% Dimethyl Sulfoxide (EuroClone), and then stored in liquid nitrogen. The lymphocyte percentages, phenotypes, and functions were evaluated by flow cytometry on thawed PBMCs. The following antibodies were used: CD3 (BV510), CD4 (APC), CD19 (APC), TCR-γδ (FITC), Vδ2 (BB700), CCR6 (PECy-7), CD161 (BB700), CD25 (BV421), CD127 (BB515), CD38 (PE), CD69 (BV421), CD86 (PECy-7), IFN-γ (PE), TNF-α (PECy-7), IL-17A (BV421), CD107a (PECy-7), and Granzyme B (PE) (BD Biosciences, Franklin Lakes, NJ, USA). Dead cells were labeled using ViobilityTM Fixable Dye (Miltenyi Biotec, Bergisch Gladbach, Germany). Combinations that were used were: CD4 + CCR6 + CD161 + (Th17); CD3 + pan-TCRγδ+ (γδ T-cells), CD4 + CD25 + CD127 − (Treg cells); CD3 + TCRγδ + CD69+/CD38 + γδ T-cell activation), CD3 + TCRγδ + GrB+/CD107a+/IFNγ+/TNFα+/IL-17A + (γδ T-cell functions); and CD3 − CD19 + CD69+/CD86 + (B-cells activation).

For surface marker staining, 1 × 10^6^ PBMCs were stained with the appropriate antibodies for 20 min at 4 °C in the dark and acquired using FACSVerse™ cytometer (BD Biosciences).

For intracellular cytokine staining, 1.5 × 10^6^ PBMCs were stimulated with phorbol myristate acetate (PMA) and ionomycin (Sigma Aldrich Merck, Burlington, MA, USA) (25 ng/mL and 1 µg/mL, respectively) for 5 h, whereas negative controls were left untreated. Brefeldin A (1 mg/mL, Sigma Aldrich) was added after 1 h stimulation. The cells were harvested and stained with surface antibodies; after paraformaldehyde (PFA) fixation (1%, Sigma-Aldrich), the cells were permeabilized with Saponin 0.2% (Sigma-Aldrich) and stained with intracellular cytokines for 30 min at room temperature (RT) and acquired using the FACSVerse™ cytometer (BD Biosciences).

An example of the gating strategy that was used is presented in Appendix A.

### 2.3. Γ-. Globulins Quantification

The quantification of γ-globulins was performed by means of serum protein electrophoresis by the central laboratory of “L. Sacco” Hospital, Milan, Italy.

### 2.4. Statistics

The data were analyzed and graphed with GraphPad Prism Software. Categorical variables are presented as the total number and percentage values. Continuous variables are presented as the median and interquartile range (IQR). A Wilcoxon matched-pairs test was applied for longitudinal analysis between T0 and SVR12 in the HIV-HCV co-infected group and in the HCV infection group. A Mann–Whitney test was applied for cross-sectional analysis of all study groups.

## 3. Results

### 3.1. Study Population

A total of 47 HIV-HCV co-infected and 35 HCV mono-infected individuals were included in the study. The demographic and clinical characteristics of the study population are summarized in Table 1. The HIV-related parameters in the HIV-HCV co-infected subjects are presented in Table 2.

The two study groups were comparable in terms of demographic and hepatic disease characteristics. All the subjects achieved SVR12 after DAA treatment.

### 3.2. Γδ T-Cell Activation and Function during DAA Treatment

No major changes were observed in the γδ T-cell percentages in the two groups, however, at SVR12, the HIV-HCV co-infected subjects showed significantly higher γδ T-cell frequencies than the HCV mono-infected individuals (2.99% [IQR: 2.81–3.22] vs. 2.34% [IQR: 1.89–2.89]; *p*-value < 0.0001) (Figure 1A).

The magnitude of activated γδ T-cells and the functionality of total γδ T-cells in the HIV-HCV co-infected and HCV mono-infected subjects were assessed prior and following DAA treatment. A significant contraction of activated γδ T-cells, i.e., CD69 + γδ T-cells and CD38 + γδ T-cells, was observed at SVR12 in individuals with HIV-HCV co-infection (CD69 + γδ +: 11% [IQR: 8.7–14]; 8.6% [IQR: 5.2–12]; *p* = 0.02; CD38 + γδ +: 55% [IQR 33.5–68]; 39% [IQR: 32.1–53]; *p* = 0.01) and HCV mono-infection (CD69 + γδ +: 10.3% [IQR: 6.7–11.6]; 6.5% [IQR: 4–8.3]; *p* = 0.01; CD38 + γδ +: 38.9% [IQR: 30.5–55.8]; 27.4% [IQR: 22–38–8]; *p* = 0.03) (Figure 1B). Of note, at SVR12, the HIV-HCV co-infected subjects displayed significantly higher CD69 + γδ + (*p* = 0.02), as well as CD38 + γδ + T-cells (*p* = 0.003) as compared to individuals with HCV mono-infection (Figure 1A,B).

We also evaluated cytokine production i.e., Granzyme B, CD107, IFN-γ, TNF-α, and IL-17A from γδ T-cells in the course of DAA treatment. With the exception of lower baseline Granzyme B + γδ+ T-cells in the co-infected individuals (29.5% [IQR: 18.6–53.2] vs. 81.8% [IQR: 30.1–93.7]; *p* = 0.01; Figure 1C), possibly suggesting functional exhaustion in this setting, no other significant difference was observed between the groups or in cytokine production change within the same group during therapy.

Similarly, Vδ2 T-cells were homogenous between the groups with a non-significant rise in the course of DAA treatment in both groups: HIV-HCV co-infected (T0: 16.9% [IQR: 4.9–24.5]; SVR 12: 13.4% [IQR: 10.5–23.06]; *p* = 0.7) and HCV mono-infected (T0: 14.4% [IQR: 7.2–3.9]; SVR 12: 18.1% [IQR: 11.5–26.8] *p* = 0.2) individuals (Figure 1D).

### 3.3. Treg and Th17-Cells during DAA Treatment

We next investigated the trend of Treg and Th17-cell frequencies during the DAA treatment in the two study groups. No differences were observed in Treg (Figure 2A) and Th17-cells, aside from a significant decrease of the latter in the HIV-HCV co-infected subjects (T0: 1.92% [IQR: 1.76–2.47]; SVR 12: 1.62% [IQR: 1.1–2.39]; *p*-0.04) following successful DAA treatment (Figure 2B).

The Vδ2/Th17 ratio increased significantly in subjects with HIV-HCV co-infection (T0: 6.3% [IQR: 2.56–12.7]; SVR 12: 11.03% [IQR: 6.04–14.24]; *p* = 0.03) and HCV mono-infection (T0: 8.63% [IQR: 5.08–10.5]; SVR12: 11.56% [IQR: 7.01–15.48]; *p* = 0.02) (Figure 2C). No statistically significant correlation was found between the Vδ2/Th17 ratio and transaminase levels as well as liver stiffness measurements at baseline in both groups (co-infection: AST: r = 0.2, *p* = 0.1; ALT: r = 0.2; *p* = 0.09; stiffness: r = 0.06, *p* = 0.7; mono-infection: AST: r = 0.1, *p* = 0.4; ALT: r = 0.1; *p* = 0.5; stiffness: r = 0.19, *p* = 0.3); however, upon successful treatment, a negative correlation was found between the Vδ2/Th17 ratio and liver function enzymes in co-infected subjects alone (AST: r = −0.3, *p* = 0.05; ALT: r = −0.4, *p* = 0.01) (Figure 2D), whereas no correlation was found in the mono-infected individuals (AST: r = −0.03, *p* = 0.8; ALT: r = 0.2, *p* = 0.2) (Figure 2E).

### 3.4. B-Cell Activation and γ-Globulins during DAA Treatment

Given that persistent Treg imbalances together with enduring B-cell activation contribute to the pathogenesis of extrahepatic manifestations of chronic HCV infection, such as cryoglobulinemic vasculitis and B-cell lymphoma [21], we also studied the magnitude of activated B-cells in the study groups prior to and following DAA treatment. A significant contraction of activated CD69 + B-cells was observed in both the HIV-HCV co-infected (4.8% [IQR: 2.7–7.3]; 3.1% [IQR: 1.5–6]; *p* = 0.02) and HCV mono-infected (2.3% [IQR: 2.1–5.2]; 2% [IQR 1.3–3.8]; *p* = 0.01) individuals. (Figure 3A). Despite these changes, significantly higher CD69-expressing B-cells were found in the HIV-HCV co-infected compared to HCV mono-infected subjects at baseline (4.9% [IQR: 2.7–7.3] vs. 2.3% [IQR: 2.1–5.2]; *p* = 0.005) and after DAA treatment (3.1% [IQR: 1.5–6] vs. 2% [IQR: 1.3–3.8]; *p* = 0.002).

A similar, albeit not statistically significant, decrease in activated CD86 + B-cells was also observed in the course of the study (Figure 3A).

Finally, a significant contraction of γ-globulin levels was observed in both the HIV-HCV co-infected (18.3% [IQR: 16.18–22.73]; 17.10% [IQR: 15–20.65]; *p* = 0.003) and HCV mono-infected individuals (19.2% [IQR: 16.53–21.8]; 17.1% [15.95–18.83]; *p* = 0.002) after DAA treatment (Figure 3B).

## 4. Discussion

Co-infection with HCV has been described as a co-factor contributing to persistent immune dysfunction in HIV-infected cART-treated, aviremic people living with HIV (PLWH) [23], which, in turn, may also influence hepatic disease progression [24].

DAAs exert a partial effect on peripheral immune imbalances in co-infected subjects; indeed, they are able to induce the phenotypic and functional recovery of HCV-specific CD8 T-lymphocytes [25] and NK-cells [26], yet not of mucosal-associated invariant T-cells (MAIT) [27]. Yet, to our knowledge, few papers have explored the effects of DAAs on key immune cell populations in fibrogenesis, namely γδ T, Treg, Th17, and B-cells [16,17,18,19], in HIV-HCV co-infected subjects [22].

In the present paper, we evaluated the kinetics of these cell subsets in a cohort of DAA-treated individuals with HIV-HCV co-infection and HCV mono-infection. Γδ T-cell activation, measured through the expression of surface markers CD38 and CD69, decreased in both groups following successful anti-HCV treatment, yet persisted at significantly higher levels in the co-infected individuals. In contrast, γδ T-cell function that was assessed by Granzyme B, CD107a, IFN-γ, TNF-α, and IL-17A production remained unchanged in both groups. Our results suggest that HCV eradication that was obtained by DAA treatment affects only γδ T-cell phenotype and not function. Indeed, despite lacking a control group of HIV mono-infected and/or healthy uninfected donors, our findings are in keeping with prior evidence by Cimini et al. who demonstrated decreased Vδ2 cell activation in the DAA-treated, HIV-HCV co-infected individuals, yet maintenance at higher levels compared to healthy donors, with little/no effect on IFN-γ production [22]. Our findings currently lack data on activated Vδ1 cells, which have been shown to play a role in hepatic disease progression in co-infected individuals regardless of cART [20]. The effect of DAA on the reported increase of Vδ1 cells in HIV-HCV co-infection is unknown and needs to be explored in this setting.

We also report a non-significant rise of Vδ2 cells in co-infected individuals, which, coupled with a decrease in Th17 frequencies, resulted in a significant increase of the Vδ2/Th17 ratio. In accordance with data demonstrating that the Vδ2/Th17 ratio is inversely linked to hepatic damage [28], we found a negative correlation between the Vδ2/Th17 ratio and liver function enzymes, albeit within the normal range, in the setting of co-infection alone. An increase in the Vδ2/Th17 ratio was also observed in mono-infected subjects but it appeared to be less consistent than that of co-infected individuals, given the lack of significant rise in Vδ2 and/or decrease in Th17-cells as well as the absence of a statistical correlation with liver transaminases. These results point to a possible differential effect of HCV clearance in HIV-HCV co-infected and HCV mono-infected subjects linked to γδ T and Th17 subsets and complement prior evidence by Farcomeni et al. showing greater differences in T-cell phenotype changes in co-infected individuals [29]. Further, in keeping with the concept that was suggested by Wu et al. [28] of the interaction between the two cell subsets in driving liver inflammation and together with the notion of low-level inflammation in treated HIV subjects [23], our findings may suggest subclinical evidence of hepatic damage in the setting of HIV-HCV co-infection.

Finally, we describe decreases in the γ-globulin concentrations following DAA-mediated SVR which may relate to the reduction in peripheral, activated B-cells. However, our data also show enduring B-cell activation in HIV-HCV co-infection possibly meaning a maintained a risk of developing extra-hepatic manifestations that are associated with prior HCV infection [21].

Our analyses were limited to the SVR12 post-DAA timing, thus a long-term effect of HCV eradication on the T- and B-cell function recovery cannot be ruled out. Indeed, elevated immune activation may persist long after the end of treatment and contribute to post-SVR consequences.

In conclusion, we show a benefit of HCV eradication that is mediated by DAA treatment in HIV-HCV and HCV mono-infected subjects, with decreased activation of γδ T- and B-cells as well as increases in the Vδ2/Th17 ratio and reduced γ-globulin concentrations. These findings shed light onto the possible immunopathogenic mechanisms underlying the improvement of liver function following HCV clearance. However, γδ T- and B-cell activation remained at higher levels in HIV-HCV co-infection than that which was measured in HCV mono-infection alone. Given the enduring clinical risk of cART-treated HIV-infected subjects [23] and the high rates of subsequent HCV re-infection in this population [30], future studies are needed to elucidate the impact of DAA therapy on long-term immune imbalances in HIV-HCV co-infected individuals.

## Figures and Tables

**Figure 1 viruses-14-01594-f001:**
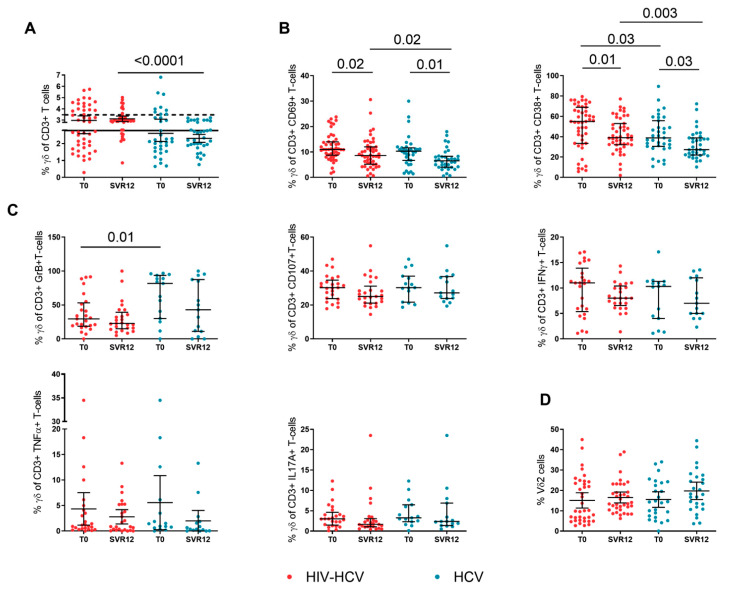
γδ T-cell phenotype, activation, function, and Vδ2 T-cells in the HIV-HCV co-infected and HCV mono-infected individuals that were undergoing DAA treatment. (**A**) Frequencies of γδ T-cells in the HIV-HCV co-infected and HCV mono-infected individuals prior and following DAA treatment. Wilcoxon matched-pairs test (*n* = 40 and *n* = 35) and Mann–Whitney test. For this experiment, the HIV mono-infected individuals (*n* = 15; bold line) and healthy controls (*n* = 10; dotted line) are also shown. The HCV mono-infected individuals showed significantly lower γδ T-cell frequencies after DAA treatment than the healthy controls (2.34% [IQR: 1.89–2.89] vs. 3.44% [IQR: 2.4–4.61] *p*-value = 0.006). (**B**) Frequencies of activated CD69+ and CD38+ γδ T-cells in the HIV-HCV co-infected and HCV mono-infected individuals prior and following DAA treatment. Wilcoxon matched-pairs test (*n* = 47 and *n* = 35) and Mann–Whitney test. (**C**) Frequencies of Granzyme B, CD107a, IFN-γ, TNF-α, and IL1–7A producing γδ T-cells in the HIV-HCV co-infected and HCV mono-infected individuals prior and following DAA treatment. Wilcoxon matched-pairs test (*n* = 25 and *n* = 15) and Mann–Whitney test. (**D**) Frequencies of Vδ2 T-cells in the HIV-HCV co-infected and HCV mono-infected individuals prior and following DAA treatment. Wilcoxon matched-pairs test (*n* = 31 and *n* = 26) and Mann–Whitney test.

**Figure 2 viruses-14-01594-f002:**
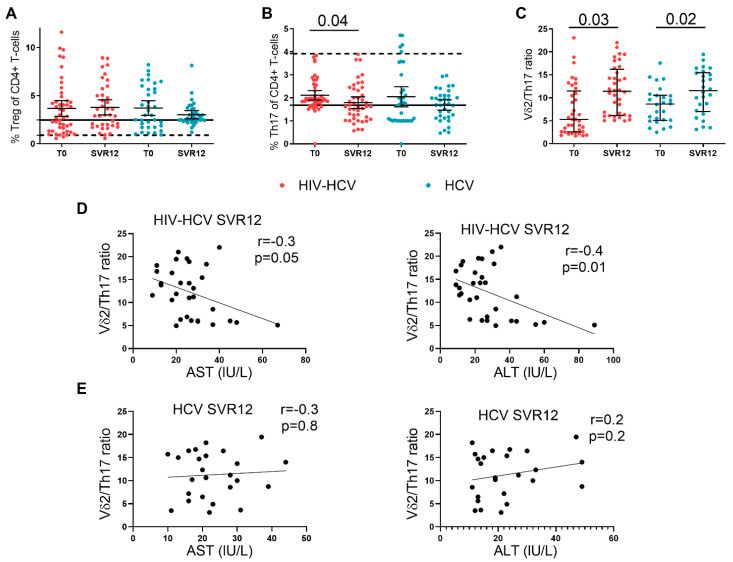
Treg, Th17, and Vδ2/Th17 ratio and correlation between Vδ2/Th17 ratio and liver function enzymes in the HIV-HCV co-infected and HCV mono-infected individuals that were undergoing DAA treatment. (**A**) Frequencies of Treg in the HIV-HCV co-infected and HCV mono-infected individuals prior and following DAA treatment. Wilcoxon matched-pairs test (*n* = 38 and *n* = 35) and Mann–Whitney test. For this experiment, HIV mono-infected individuals (*n* = 15; bold line) and healthy controls (*n* = 10; dotted line) are also shown. The HCV mono-infected individuals showed significantly higher Treg cell frequencies than the healthy controls both before and after DAA treatment (T0: 2.3% [IQR: 1.7–5.7] vs. 0.82% [IQR: 0.62–1.95]; *p*-value = 0.0001; SVR 12: 3% [IQR: 1.87–5.22] vs. 0.82% [IQR: 0.62–1.95]; *p* = 0.0002) (**B**) The frequencies of Th17 in the HIV-HCV co-infected and HCV mono-infected individuals prior and following DAA treatment. Wilcoxon matched-pairs test (*n* = 43 and *n* = 35) and Mann–Whitney test. For this experiment, the HIV mono-infected individuals (*n* = 15; bold line) and healthy controls (*n* = 10; dotted line) are also shown. HCV mono-infected individuals showed significantly lower Th17-cell frequencies than the healthy controls both before and after DAA treatment (T0: 1.75% [IQR: 1–3] vs. 3.92% [IQR: 3.38–5.62] *p*-value = 0.002; SVR 12: 1.71% [IQR: 1.17–2.1] vs. 3.92% [IQR: 3.38–5.62]; *p* < 0.0001) (**C**) The Vδ2/Th17 ratio in HIV-HCV co-infected and HCV mono-infected individuals prior and following DAA treatment. Wilcoxon matched-pairs test (*n* = 31 and *n* = 26) and Mann–Whitney test. (**D**) Spearman’s correlation between Vδ2/Th17 ratio and aspartate aminotransferase (AST) and alanine aminotransferase (ALT) in 31 HIV-HCV co-infected individuals after DAA treatment. (**E**) Spearman’s correlation between Vδ2/Th17 ratio and aspartate aminotransferase (AST) and alanine aminotransferase (ALT) in 25 HCV mono-infected individuals after DAA treatment.

**Figure 3 viruses-14-01594-f003:**
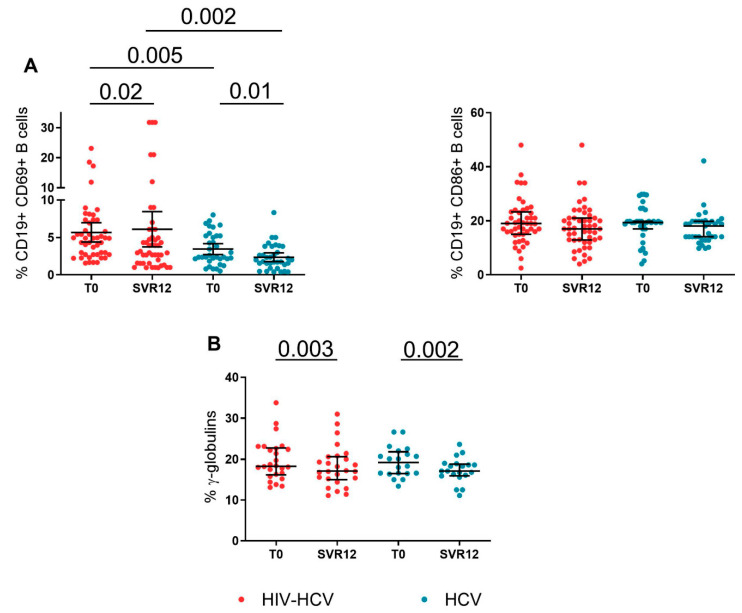
B-cell activation and γ-globulins in the HIV-HCV co-infected and HCV mono-infected individuals that were undergoing DAA treatment. (**A**) Frequencies of activated CD69+ and CD86+ B-cells in the HIV-HCV co-infected and HCV mono-infected individuals prior and following DAA treatment. Wilcoxon matched-pairs test (*n* = 47 and *n* = 35) and Mann–Whitney test. (**B**) γ-globulins in the HIV-HCV co-infected and HCV mono-infected individuals prior and following DAA treatment. Wilcoxon matched-pairs test (*n* = 26 and *n* = 20) and Mann–Whitney test.

**Table 1 viruses-14-01594-t001:** Demographic and clinical characteristic of the study subjects.

	HIV-HCV	HCV	*p*-Value
	(N = 47)	(N = 35)	
Male, *n* (%)	37 (79)	22 (63)	0.13
Age (years), median (IQR)	52 (48–57)	54 (47–62)	0.66
Time since HCV diagnosis, years (IQR)	12 (4–23)	14 (2–20)	0.22
HCV genotype, *n* (%)			0.8
1	23 (49)	17 (48)	
2	2 (4)	5 (14)	
3	15 (32)	4 (11)	
4	7 (15)	8 (23)	
5	0	1 (3)	
Liver fibrosis			0.7
Stiffness < 9.5 KPa (F0–F2)	35 (74)	28 (80)	
Stiffness ≥ 9.5 KPa (F3–F4)	11 (23)	7 (20)	
DAA regimen, *n* (%)			0.02
ombitasvir/paritaprevir/rtv	1 (2.1)	2 (5.7)	
ombitasvir/paritaprevir/rtv + dasabuvir	2 (4.2)	3 (8.6)	
glecaprevir/pibrentasvir	9 (19.1)	17 (48.6)	
ledipasvir/sofusbuvir + rbv	4 (3.5)	0 (0)	
sofusbuvir/daclatasvir	6 (12.8)	2 (5.7)	
sofusbuvir/velpatasvir	18 (38.3)	5 (14.3)	
elbasvir/grazoprevir	5 (10.7)	5 (14.3)	
ombitasvir + rbv	2 (4.2)	0 (0)	
sofusbuvir/daclatasvir + rbv	0 (0)	1 (2.9)	
DAA duration, *n* (%)			0.2
8 weeks	9 (19)	12 (34)	
12 weeks	34 (72)	21 (30)	
24 weeks	4 (8)	2 (6)	
Liver transaminases at SVR12			
AST, IU/L (median, IQR)	25 (20–30)	21 (17–28)	0.2
ALT, IU/L (median, IQR)	24 (15–32)	20 (13–26)	0.2

Legend: Liver fibrosis assessed by elastography; DAA: Direct Acting Antivirals; rtv: ritonavir; rbv: ribavirin.

**Table 2 viruses-14-01594-t002:** HIV-related parameters in the HIV-HCV co-infected subjects.

	HIV/HCV
	(N = 47)
Time since HIV diagnosis, years (IQR)	17 (11–25)
CD4% at T0, median (IQR)	28 (22–34)
CD4, cell/mmc at T0, (IQR)	612 (327–831)
CD4/CD8 ratio, median (IQR)	0.66 (0.49–0.95)
Patients with HIV/RNA < 40 cp/mL, *n* (%)	47 (100)

## Data Availability

Data that support the findings of this study are available upon reasonable request to the corresponding author.

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
