# Peer review of "Gamma-Delta T-Cell Phenotype and Function in DAA-Treated HIV-HCV Co-Infected and HCV-Mono-Infected Subjects"

_viruses, 2022, doi:10.3390/v14081594_

Round 1

Reviewer 1 Report

The paper of Bono V et all focused on DAA therapy impact on some immune parameters, that were reported to impact hepatic disease progression, in HIV/HCV co-infected as compared to HCV mono-infected subjects. By this paper the authors try to elucidate the impact of DAA on some immune cell populations involved in fibrogenesis.  They have found significant differences in gamma/ delta T cells and B cells activation after HCV cure in both groups. They found that the ratio between Vδ2 and Th17 cells inversely correlated with liver transaminases level. However the importance of these results is poorly discussed, moreover HIV impact is not discussed.

 Major points:

The paper show only Vδ2 of γδ T cells and no data on Vδ1, which was previously reported to be increases in the circulation HIV /HCV coinfected subjects and contribute to liver disease progression. (AIDS Res Hum Retroviruses (2001) 17(14):1357–63.); HIV infection induces a Vδ1 mobilization from tissue to the peripheral blood and depletion of Vδ2. I think if you want to have a better image for liver damage in HIV/HCV co-infection, this population should be evaluated. 

Another major point is the correlation between transaminases level at SVR12 and Vδ2/Th17 ratios. The authors suggests that is negatively linked to liver damage; however the AST and ALT levels at SVR 12 (table 1) seems to be in normal range, they should discuss more on this.

The flow cytometry gating strategy should be included in the figures (at least as supplemental material if not allowed by journal guidelines).

The conclusions are not supported by the results: 

lane 25-27: "DAA had a limited effect on the phenotype and function of γδ T cells, frequency of Treg and Th17 cells as well as B cells activation in HIV-HCV co-infection."  - only the statistically significant results should be considered 

lane 238-242: “In conclusion, we show a partial benefit of HCV eradication mediated by DAA treatment on the phenotype and function of γδ T cells as well as Treg and Th17 frequencies in the setting of HIV-HCV co-infection. These findings shed light onto the possible immunopathogenic mechanisms underlying the improvement of liver function following HCV clearance.” - same here

In addition the differences between groups, such as the grB+γδ T cells, should be given a proper discussion.

Minor:

In the methods section for B cell activation markers should be CD3-CD19+CD69+/CD86+

It would be nice to have the Figures as dots not bars. In the axes title (or in the figure legend) should be noted to which cell population is the percentage reported (ex: %  γδ T cells of CD3+ T cells)

Check the number of subjects to be consistent, at figure 1D (36 HIV/HCV si 29 HCV) at figure 2C (36 HIV/HCV si 30 HCV)

Other points:

Are there any correlations at baseline with liver damage (Fibrosis, ALT, AST). - if performed should be mentioned at least in the discussion

Reviewer 2 Report

           The manuscript entitled “Gamma-delta T-cell Phenotype and Function in DAA-Treated HIV-HCV Coinfected and HCV-Monoinfected Subjects” by Bono. V et.al. demonstrated the effect of DAA treatment on the Gamma-delta T cells phenotype and functions in the HCV monoinfected and HIV-HCV coinfected cohorts and reported a decreased activation of gamma-delta T cells in both the cohorts however, the overall frequency was higher in coinfected individuals. They further reported no effect on the functionality of these cells with SVR. It is of great interest to explore the functions of these cells. I have a few suggestions though.

1.  My suggestion is to include the HIV monoinfected and healthy uninfected individuals as a control.

2.  The significance of the study is not very clear from the discussion, the author should improve the discussion by highlighting the importance of the study.

3.  I suggest replacing the bar graphs with scatter dot plots showing the individual values of each of the phenotypic and functional markers.

4. Also include the flow cytometry representative plots in the figures demonstrating the differences in the groups.

Round 2

Reviewer 1 Report

The current form of the paper was improved and I consider is suitable for publication

Reviewer 2 Report

My concerns are addresse